Review
# Conserved 3D genome reorganization during DNA repair

Veysel Oğulcan Kaya[1] , Ogün Adebali[1] , Hanspeter Naegeli[2] , Michelle N Yancoskie[2]

The mammalian genome is hierarchically structured to maintain accessibility and flexibility for essential nuclear processes such as transcription and DNA repair. A recent study using high-throughput 3D genome mapping reveals that nucleotide excision repair triggers large-scale chromatin rearrangements, reinforcing topologically associating domains and chromatin compartments. Notably, similar principles of chromatin stabilization are observed during double-strand break repair, consistent with the notion that chromatin restructuring may be an active and conserved DNA repair strategy rather than a passive consequence of damage. The observed stabilization is postulated to optimize repair efficiency by reducing the search space for damage, enhancing DNA accessibility at damage sites, increasing local concentrations of repair factors, and preventing aberrant chromosomal rearrangements. By synthesizing emerging evidence on nucleotide excision repair–driven chromatin dynamics and its parallels with double-strand break repair, this review examines how genome architecture actively shapes the DNA damage response and highlights broader implications for genetic diseases and therapeutic strategies.

## Introduction

The genome must balance stability with chromatin accessibility to enable critical processes such as transcription and DNA repair. Over the past two decades, chromosome conformation capture techniques such as high-throughput, genome-wide Hi-C have revealed the hierarchical nature of genome organization, from chromosome territories (Fig 1A) and compartments (Fig 1B) to topologically associating domains (TADs; Fig 1C) and chromatin loops (Fig 1D) (Denker & de Laat, 2016). The "fractal globule" (Grosberg et al, 1993) or "crumpled polymer" model describes chromatin as a densely packaged yet unknotted, easily accessible polymer (Mirny, 2011; Goundaroulis et al, 2020; Polovnikov et al, 2023; Hildebrand et al, 2024) whereby globule-like units repeat across scales: chromatin loops cluster into sub-TADs, and sub-TADs into TADs. This flexible arrangement avoids entanglement

while maintaining the ability to rapidly reconfigure. For instance, during transcription, chromatin loops dynamically form between enhancers and their target promoters to facilitate gene expression, in a highly cell type–specific manner and often over long genomic distances (Rao et al, 2014). The importance of this spatial organization manifests in the fact that disruptions to key chromatin architectural players such as CTCF, cohesin, and the cohesin loader NIPBL are implicated in severe developmental disorders (Krantz et al, 2004; Deardorff et al, 2012; Valverde de Morales et al, 2023).

Genome organization has heretofore been studied largely in the context of transcription. However, chromatin restructuring is a critical event during numerous additional processes, including DNA recombination (Peters, 2021), replication (Emerson et al, 2022), and repair. DNA repair pathways comprise the DNA damage response (DDR), which maintains genome integrity by detecting and resolving various genotoxic threats (Rouse & Jackson, 2002). A key function of the chromatin restructuring observed during the DDR is to enable repair machinery to stably bind to chromatin to access damage sites. At a local level, posttranslational modifications (PTMs) of histones, such as acetylation and methylation, regulate chromatin compaction and stability, and ATP-dependent chromatin remodeling complexes reposition or even evict histones. The extensive body of research on these nucleosome-centric mechanisms is reviewed elsewhere (Polo & Almouzni, 2015; Apelt et al, 2021; Song et al, 2023).

Recent technological advances, especially the increasing resolution and availability of Hi-C and damage-mapping techniques, have enabled genome-wide tracking of how chromatin architecture responds to DNA damage. Multiple recent studies (Collins et al, 2020; Sanders et al, 2020; Arnould et al, 2021; Zagelbaum et al, 2023; de Luca et al, 2024; Kaya & Adebali, 2025) have demonstrated that at least two DNA repair pathways induce large-scale reorganization of the mammalian genome, consistent with the idea that genome architecture bolsters repair efficiency by creating a spatially optimized environment for damage recognition and resolution. Given the growing body of work linking genome organization to the DDR, it is timely to synthesize insights into their interplay, providing a baseline expectation for future studies that investigate additional DNA-damaging agents and repair pathways. In this review, we examine emerging evidence that mammalian genome structure plays an active role in the DDR. We begin by describing how the highly dynamic nature of the 3D genome

[1]The Molecular Biology, Genetics, and Bioengineering Program, Faculty of Engineering and Natural Sciences, Sabancı University, Istanbul, Türkiye [2]Institute of Veterinary Pharmacology and Toxicology, Vetsuisse Faculty, University of Zurich, Zurich, Switzerland

Correspondence: myancoskie@vetpharm.uzh.ch

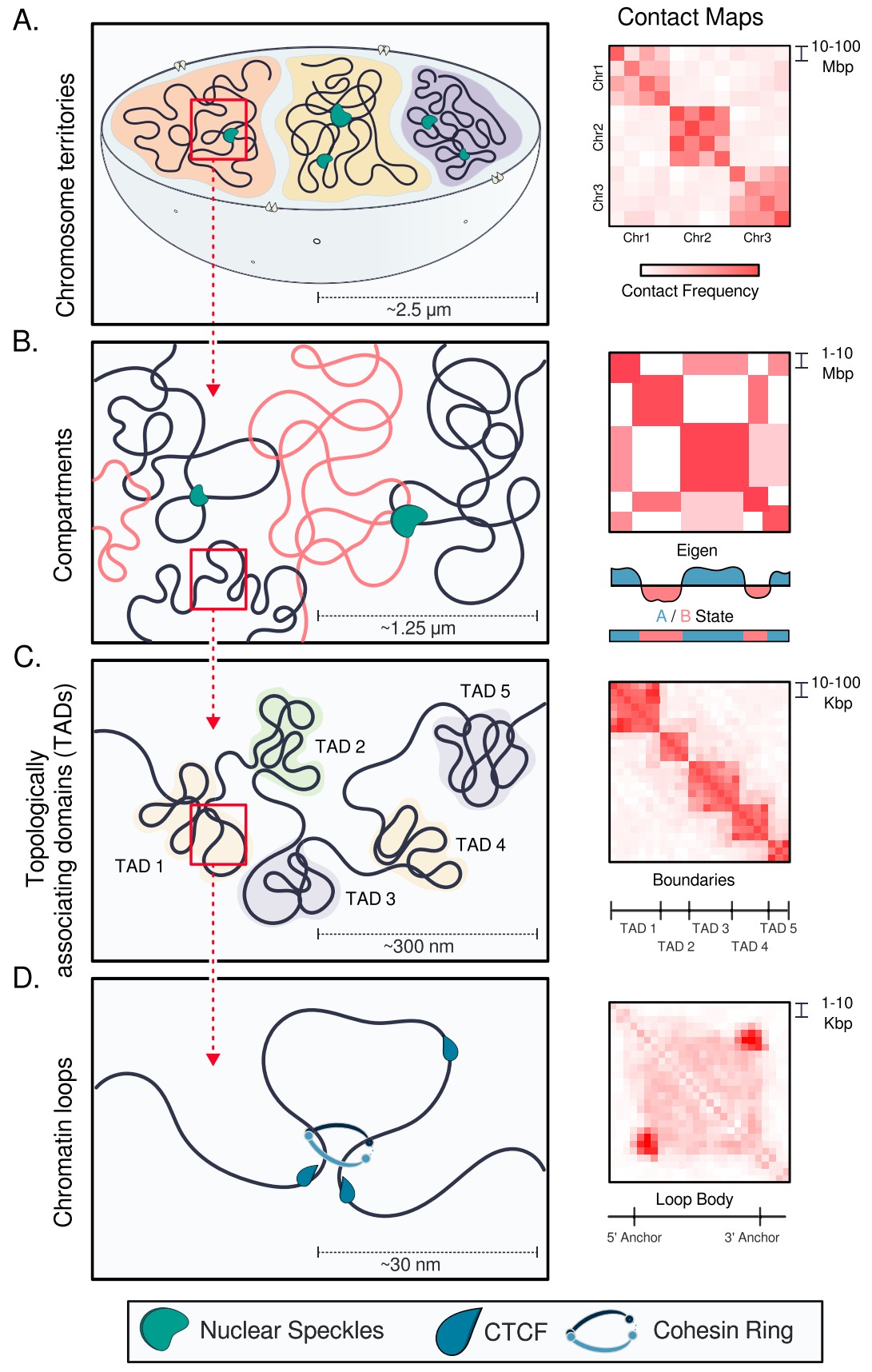

A. Chromosome territories

Contact Maps

~2.5 μm

10-100 Mbp

Contact Frequency

B. Compartments

~1.25 μm

1-10 Mbp

Eigen

A / B State

C. Topologically associating domains (TADs)

TAD 5
TAD 2
TAD 1
TAD 3
TAD 4

~300 nm

10-100 Kbp

Boundaries

TAD 1  TAD 3  TAD 5
TAD 2  TAD 4

D. Chromatin loops

~30 nm

1-10 Kbp

Loop Body

5' Anchor    3' Anchor

Nuclear Speckles    CTCF    Cohesin Ring

facilitates nuclear processes and examine the challenges that have historically impeded the mapping of these chromatin dynamics during the DDR. We review recent studies that have overcome these obstacles, with a focus on double-strand break repair (DSBR), which resolves one of the most abundant and severe forms of DNA damage, and extrapolate to nucleotide excision repair (NER), which resolves helix-distorting DNA lesions such as those caused by UV radiation. In light of the potential of harnessing 3D genome architecture to optimize DNA-targeting therapeutic strategies, especially during cancer treatment (Lomberk & Urrutia, 2015; Tan et al, 2024 Preprint), we conclude by considering the broader implications of these findings for future research, especially for genetic diseases associated with defects in architectural proteins.

# Adaptability of the 3D genome

The dynamic nature of the 3D genome is especially striking during cell division. Mitotic entry is marked by a dramatic but reversible global rearrangement of chromatin. Loops, TADs, and compartments completely disappear from chromatin contact maps, and nuclear structures such as the nucleolus, nuclear envelope, and nuclear speckles dissolve as the genome drastically reorganizes into an intrachromosomally entangled, axially compressed, bottlebrush-shaped helical array (Naumova et al, 2013; Rai et al, 2018; Abramo et al, 2019; Hildebrand et al, 2024). The mitotic loss of TADs and loops is attributable to the actions of condensins, a type of structural maintenance of chromosomes (SMC) complex. Condensin II compacts the chromatin into large loops, whereupon condensin I forms nested smaller loops within the large loops (Walther et al, 2018). Starting in telophase, the interphase chromatin structures quickly and completely reform as cohesin, another type of the SMC complex, takes over from condensin (Abramo et al, 2019; Kang et al, 2020; Hildebrand et al, 2024). Topoisomerase II, which is active throughout the cell cycle, keeps chromatin unentangled during interphase. These studies reveal a high inherent flexibility adaptable to biological processes. Two main mechanisms thought to underlie chromatin structure, phase separation, and chromatin loop extrusion permit this large degree of genome elasticity. Insights into these mechanisms have largely come from chromosome conformation capture (3C)–based technologies such as Hi-C (Lieberman-Aiden et al, 2009) and Micro-C

(Hsieh et al, 2015), which cross-link chromatin, digest and religate interacting DNA fragments, and sequence the resulting chimeric junctions to infer contact probabilities; and from complementary imaging approaches that visualize the physical proximity of genomic loci in single cells (Denker & de Laat, 2016; Fig 1).

## Phase separation

Imaging techniques such as fluorescence in situ hybridization and live-cell imaging have implicated phase separation in shaping chromatin at large scales (i.e., chromosome territory and chromatin compartment-level resolution) (reviewed in Li et al [2023] and Ng et al [2022]). Originally conceptualized in polymer physics, phase separation underlies the formation of membraneless nuclear bodies such as the nucleolus and nuclear speckles. Two forms of phase separation feature prevalently in chromatin biology: liquid–liquid (LLPS) and polymer–polymer phase separation (PPPS). LLPS segregates euchromatin from heterochromatin, as well as facultative from constitutive heterochromatin. During LLPS, weak, multivalent molecular interactions between nucleic acids and proteins—especially those proteins containing intrinsically disordered regions, whose lack of rigid structural constraints allows flexible interactions with multiple other molecules—cause chromatin and its associated proteins to assemble into liquid droplets. For example, within constitutive heterochromatin, heterochromatin protein 1 (HP1) and repressive histone H3K9 methylation marks aggregate into liquid droplets that sequester methylated from active chromatin (Li et al, 2023). These molecular interactions, despite their transience, establish chromatin compartments that are relatively stable on a cellular timescale; for example, HP1 rapidly binds and unbinds chromatin within seconds, yet establishes stable repressive chromatin states that persist for hours (Akilli et al, 2024). During PPPS, chromatin acts as a polymer scaffold, bridging distant genomic regions through nonspecific interactions between chromatin-associated proteins and chromatin modifications (Ng et al, 2022).

Phase separation influences the kinetics and specificity of nuclear transactions by sequestering and concentrating reaction components within the same spaces (reviewed by Akilli et al [2024]). For example, transcription factors are concentrated at super-enhancers ("transcriptional condensates") during transcription, and repair proteins at damage sites ("repair foci") during the DDR (Ng et al, 2022). More globally, the chromatin compartments shaped by phase separation, despite their relative stability,

**Figure 1. Fractals of the mammalian genome.**
**(A)** Interphase chromosomes occupy distinct territories whose transcriptional activity inversely correlates with distance from the nuclear center (reviewed in Cremer and Cremer [2010]). **(B)** At megabase pair (Mbp) resolution, interphase chromatin segregates into two compartment types termed A and B. Chromatin in Compartment A (red) tends to be euchromatic, gene-rich, and transcriptionally active; in B (blue), heterochromatic, gene-poor, and transcriptionally repressed. Compartments are inferred by assigning eigenvector values to long-range (around 1-Mbp resolution) chromatin interactions captured by Hi-C or Micro-C. Intracompartment interactions manifest as plaid-like patterns on contact probability maps (heatmaps right of chromatin schematics) (Lieberman-Aiden et al, 2009; Rao et al, 2014; Hsieh et al, 2015). Nuclear speckles, membraneless nuclear bodies that function in pre-mRNA splicing and gene regulation, associate with Compartment A to help maintain intracompartment interactions (Tammer et al, 2022; Bhat et al, 2024). **(C)** At 10- to 100-kilobase pair (kbp) resolution, each interphase chromatin compartment subdivides into topologically associating domains (TADs) that constitute neighborhoods of interacting loci. TADs, which manifest as triangles along the diagonal of the contact map, span up to 2 Mbp in mammals, with interactions occurring more frequently within than across TADs (Lieberman-Aiden et al, 2009; Dixon et al, 2012; Nora et al, 2012). Nested within TADs are sub-TADs (smaller triangles). **(D)** At 5- to 10-kbp resolution, points or dots and stripes or flames visible at the corners and edges of TADs can be explained through loop extrusion, whereby cohesin (tripartite ring) extrudes DNA until halted by a boundary element such as CCCTC-binding factor (CTCF), forming a chromatin loop that brings distal loci into spatial proximity (Rao et al, 2014; Fudenberg et al, 2016; Wang et al, 2016).

can shift to facilitate nuclear processes. For example, after double-strand break (DSB) induction, Hi-C has revealed compartment transitions biased toward active chromatin Compartment A, potentially facilitating access to damage sites (Zagelbaum et al, 2023).

### Loop extrusion

At the TAD and sub-TAD level, where TADs are defined as self-interacting, megabase pair-scale regions, the genome is organized by loop extrusion. The ring-shaped, ATP-dependent SMC complexes act as loop-extruding factors by binding two loci simultaneously and reeling the intervening DNA into progressively larger loops as they translocate along the DNA (Sanborn et al, 2015; Fudenberg et al, 2016; Wang et al, 2016). SMC complexes include SMC5/6 and the aforementioned condensins and cohesins. Each comprises two SMC subunits (SMC1/3 in cohesin, SMC2/4 in condensin, and SMC5/6) that heterodimerize through their hinge domains and a kleisin subunit (RAD21, for radiation-sensitive 21, in cohesin; CAPH or CAPH2 in condensin I or II, respectively; NSE2 in SMC5/6) into a tripartite ring (reviewed in Kabirova et al [2023]). SMC5/6 is active in DNA repair, whereas cohesin and condensins I and II coordinate their activities to drive the assembly, maintenance, and disassembly of interphase and mitotic chromatin (Kabirova et al, 2023; Hildebrand et al, 2024). Loop extrusion is proposed to occur one- or two-sidedly depending on whether one or both of the SMC motor subunits engage in reeling the DNA, with one-sided looping occurring when one subunit remains anchored (Banigan et al, 2020). Loops from pairwise chromatin interaction data are commonly erroneously portrayed as flower-shaped, synergistic hubs that centralize nuclear processes through the accumulation of, for example, transcription factors. However, long-range reads capturing multi-way interactions reveal that loops instead follow nonrandom, ordered paths as the loop extruder progresses along the DNA (Olivares-Chauvet et al, 2016; Tavares-Cadete et al, 2020).

Loop extrusion is highly dynamic. Mammalian cohesin extrudes hundreds of base pairs of DNA per second (Davidson et al, 2019; Kim et al, 2019), and cohesin-mediated loops persist for only a few minutes (Gabriele et al, 2022; Mach et al, 2022). Factors affecting loop extrusion dynamicity, such as SMC residency time, binding specificity (although both cohesin and condensin are loaded preferentially onto AT-rich sequences) (Kim et al, 2019), and behavior upon encountering other SMC complexes on the chromatin, remain largely undescribed in mammals.

Loop extrusion halts when a loop-extruding factor stalls, is unloaded from, or dissociates from the DNA. The most prominent loop extrusion-halting "boundary element" in the mammalian genome is the stationary barrier posed by the zinc-finger protein CCCTC-binding factor (CTCF). TAD boundaries on Hi-C maps are highly enriched for CTCF-bound sites, supporting a role in blocking inter-TAD interactions (Dixon et al, 2012). CTCF bestows boundary insulation in a dosage-dependent manner whereby multiple adjacent CTCF-bound sites confer strong insulation against inter-TAD interactions (Kentepozidou et al, 2020; Amândio et al, 2021; Huang et al, 2021; Chang et al, 2023). CTCF-bound sites at each end of a TAD are usually convergently oriented—that is, pointing toward one another such that their N termini face the inside of the TAD (Rao et al, 2014; Tang et al, 2015). CTCF in addition contributes to loop extrusion within TADs (Amândio et al, 2021; Gabriele et al, 2022). By an incompletely understood mechanism, the N terminus but not C terminus of CTCF blocks in vitro cohesin progression on either or both sides of an extruded loop in a manner dependent on local DNA tension (Davidson et al, 2023; Zhang et al, 2023). When in vitro CTCF is C-terminally oriented with respect to the extruding cohesin, it can switch the direction in which the latter translocates (Davidson et al, 2023). Thus, CTCF, termed the "master weaver of the genome" (Phillips & Corces, 2009), regulates cohesin behavior even when not acting as a boundary element.

# Genome elasticity during the DDR

### Challenges of damage and repair mapping

Tracking chromatin restructuring during the DDR remains challenging because of the highly complex, interrelated nature of DDRs (Fig 2) and the limitations of current DNA damage and repair mapping methods. Numerous genome-wide techniques have been developed to capture specific types of DNA damage and repair events. For NER-processed lesions, enzyme- and antibody-based approaches such as damage-seq, CPD-seq, and XR-seq map UV radiation-induced photoproducts and excision events. Other assays detect oxidative or alkylative damage—for example, click-code-seq for 8-oxodG or abasic sites. Single-strand breaks (SSBs), which trigger base excision repair, can be profiled using ligation-based methods such as SSB-seq or GLOE-seq, whereas DSBs are mapped by methods such as BLESS, END-seq, or DSBCapture, which tag free DNA ends with biotinylated adapters (reviewed in Mingard et al [2020]). These sequencing-based approaches, however, are subject to biases such as sequence composition, chromatin accessibility including minor and major groove positioning, and DNA strandedness (Zavala et al, 2014; Heilbrun et al, 2021).

Beyond these technical biases, a separate challenge arises from the inherently stochastic nature of damage formation whereby lesions occur at different genomic positions in every cell within a population. This heterogeneity complicates the interpretation of population-averaged damage maps. Artificial systems such as those that introduce DSBs at defined loci via inducible restriction enzymes (Iacovoni et al, 2010) have been designed to overcome this issue and allow controlled mapping around uniform damage sites. Single-cell approaches can also help resolve the heterogeneity in damage formation and repair. However, such approaches are still rare. Instead, repair (Hu et al, 2019)—which can also be thought of as the disappearance of damage over time (Hu et al, 2017)—can be inferred by dividing the genome into bins of different lengths and computing the signal per bin from multiple cells. This binning approach limits the resolution at which damage can be detected, but is needed to compensate for the typical sparsity of genome-wide damage data (Elliott et al, 2023). Furthermore, the interpretation of damage and repair maps is not always straightforward. Some agents continue to generate new damage after the initial damage has been resolved, resulting in overlapping damage and repair profiles. For example, exposing

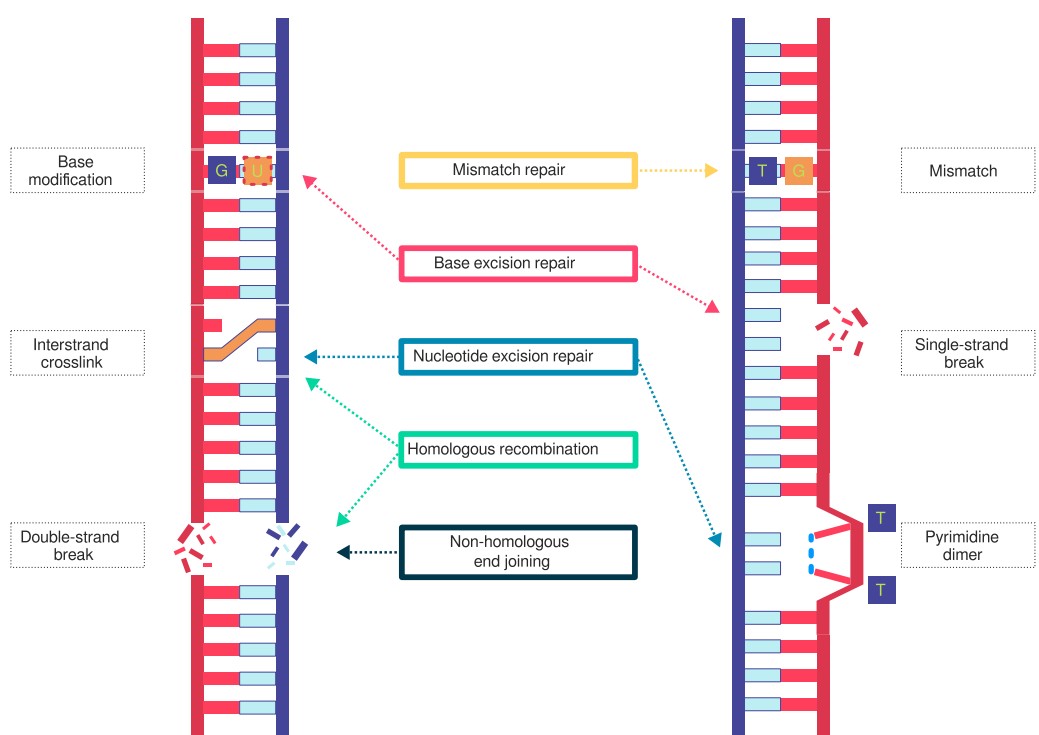

**Figure 2. DNA damage and repair overview.**
The mammalian genome must contend at all times with myriad exogenous and endogenous threats to its integrity. A single stressor can induce multiple types of DNA damage, which in turn can trigger multiple repair pathways. Conversely, a single type of damage can engage multiple repair pathways. Several major types of DNA damage and their specialized but partly overlapping repair pathways are depicted. Mismatch repair addresses base mismatches, which arise from replication errors. Base excision repair resolves single-strand breaks, which form from reactive oxygen species, other genotoxic agents, or as intermediates of repair and replication. Homologous recombination, nucleotide excision repair, and translesion synthesis coordinate to fix interstrand cross-links. Nucleotide excision repair resolves helix-distorting UV photoproducts such as cyclobutane pyrimidine dimers and pyrimidine (6-4) pyrimidone photoproducts, as well as bulky adducts (not depicted) including intrastrand cross-links, DNA–protein cross-links, aromatic adducts, and cyclopurine lesions. Smaller, less helix-distorting base modifications, including deamination, methylation, or other chemical alterations to bases, are resolved through base excision repair or (not depicted) directly reversed by enzymes. Double-strand breaks, from unresolved single-strand breaks, reactive oxygen species, irradiation, or programmed events during meiosis or immune cell diversification, are repaired through homologous recombination or nonhomologous end joining.

melanocytes to UV-A radiation results in the formation of "dark" cyclobutane pyrimidine dimers (CPDs), which, through the generation of reactive oxygen species, continue to newly form long after the exposure has ended, accounting for over half of the total UV-induced CPDs detected in these cells (Premi et al, 2015). Similarly, certain DNA-alkylating agents, such as acylfulvenes, have highly complex damage formation and repair kinetics owing to their cytosolic activation requirements and their slow translocation into the cell nucleus (Jaspers et al, 2002). Finally, most genome-wide techniques that capture DNA damage or repair give only a partial view of the DDR because they are limited to capturing only one type of damage at a time.

The high complexity of the DDR and of damage formation patterns raises the question as to how repair efficiency is achieved. CPDs, one of the chief photoproducts induced by UV radiation, typify the need for efficient repair. A typical UV exposure from sunlight can induce hundreds of thousands of CPDs per skin cell (Mouret et al, 2006). The resulting disruption of the double helix is so minimal as to require a specialized recognition factor, DDB2, to detect the CPDs (Rapić-Otrin et al, 2003). The global-genome NER (GG-NER) machinery must scan the six billion base pairs of the human diploid genome to find myriad lesions, all while avoiding collisions with transcriptional machinery. The 3D genome likely plays a pivotal role in achieving repair efficiency, much like its established role in coordinating transcription. Despite our lack of a deep mechanistic understanding of the relationship between large-scale chromatin structure and various DDRs, compelling links between the DDR and chromatin structure have long been established.

## Mechanistic links between chromatin structure and DNA repair from nonsequencing approaches

Early non–sequencing-based observations of chromatin behavior after DNA damage led to several conceptual models linking nuclear organization to genome protection and repair. In 1975, the "bodyguard hypothesis" was proposed stipulating that heterochromatin absorption of DNA damage at the nuclear periphery protects euchromatin at the nuclear center (Hsu, 1975). This hypothesis is consistent with later NER studies that used DNA damage sequencing to map UV radiation–induced photoproducts to the genome while considering their position with respect to the

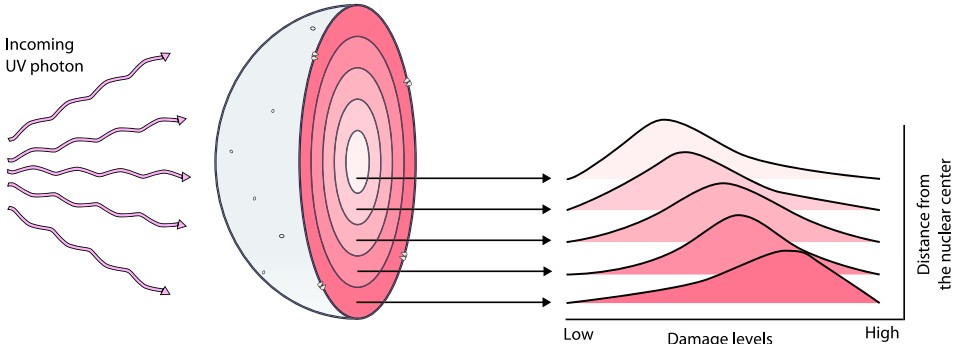

**Figure 3. Spatial positioning of chromatin within the nucleus influences UV radiation–induced damage and repair.** Genome-wide damage-mapping techniques that capture the formation and repair of UV-induced DNA lesions, such as cyclobutane pyrimidine dimers (CPDs) and pyrimidine (6-4) pyrimidone photoproducts (6-4 PPs), reveal spatial patterns reminiscent of the classic "bodyguard hypothesis" (Hsu, 1975). Sequence composition is a major determinant of UV lesion frequency, with both lesion types forming prevalently at TT dinucleotides, and TC sites more enriched in 6-4 PPs than CPDs (Hu et al, 2015). After correcting for sequence biases through simulation-based normalization, Akköse and Adebali (2023) found that both CPDs and 6-4 PPs occur more frequently than expected in peripheral chromatin (Akköse & Adebali, 2023). Across studies, genome-wide, base-corrected maps of UV lesions show that lamina-associated, peripherally located chromatin accumulates more initial UV lesions and repairs them more slowly (García-Nieto et al, 2017; Perez et al, 2021). These results are consistent with a functional shielding effect in which 3D nuclear architecture modulates both damage susceptibility and repair efficiency. This spatial arrangement likely evolved under the selective pressure imposed by UV-induced lesions, favoring cells that could minimize mutagenic damage in essential genomic regions. Over time, the evolutionary demand to mitigate the detrimental effects of UV damage may have driven the development of nuclear architectures that position more vulnerable or critical genomic regions in less exposed locations (Akköse & Adebali, 2023). This reflects a broader adaptive strategy whereby genome organization itself serves as a shield against environmental genotoxic stressors, promoting cell survival and genome integrity.

nuclear periphery. After correcting for sequence bias, a major determinant of UV lesion formation frequency, with simulation-based normalization, Akköse and Adebali (2023) found that peripheral, heterochromatic genomic regions sustain more damage than expected from sequence composition alone (Akköse & Adebali, 2023). These analyses are consistent with other reports that lamina-associated, peripherally located chromatin accumulates more initial UV lesions and repairs more slowly than interior chromatin (García-Nieto et al, 2017; Perez et al, 2021) and extend the bodyguard hypothesis: chromatin compaction and radial positioning jointly influence damage formation and repair (Fig 3). Concomitantly, the "access–repair–restore" model is built on the observation that UV radiation–induced repair is accompanied by nucleosome rearrangements. It postulated that chromatin undergoes restructuring, especially nucleosome repositioning, that permits access to damage by the repair machinery, with restoration of original structure after repair completion (reviewed in Polo and Almouzni [2015] and Smerdon [1991]).

Parallels between transcription, repair pathways, and their molecular components point to the DDR potentially reusing the same units of chromatin that feature in transcription. For example, the transcription-coupled NER (TC-NER) pathway is inherently linked to transcription because it relies on a stalled RNA polymerase to signal damage on the transcribed strand. Moreover, NER relies on key factors shared with transcription, most notably the transcription initiator complex TFIIH, the active promoter-associated histone mark H3K4me3, and the transcription start site–associated histone chaperone FACT (Balbo Pogliano et al, 2017; Maritz et al, 2023).

More directly, the DDR is linked to the 3D genome through the recurrent recruitment of chromatin architectural proteins to damage sites and their direct interactions with DDR factors. Interactions between SMC complexes and repair machinery are observed across multiple DDR pathways, wherein they tend to enhance repair: condensin I cooperates with base excision repair machinery to resolve SSBs (Heale et al, 2006; Kong et al, 2011);

SMC5/6 promotes repair of replication-created DSBs (Pond et al, 2019); cohesin, its loader NIPBL, and CTCF localize to and facilitate repair of laser- or ionizing radiation–induced DSBs, with CTCF depletion incurring increased radiosensitivity (Bauerschmidt et al, 2010; Bot et al, 2017; Sanders et al, 2020; Mamberti et al, 2022); and cohesin, NIPBL, and the chromatin remodeler RSF1 localize to UV-induced photolesion sites in immortalized human fibroblasts (Stefos et al, 2021). Interactions with CTCF likewise feature during the DDR: its association with the SSB and NER factor Cockayne syndrome B (CSB) protein strengthens upon oxidative damage (Boetefuer et al, 2018; Lake et al, 2022); it is recruited by NER endonucleases XPF and XPG to form chromatin loops at nuclear receptor target genes (Le May et al, 2012); and it cooperates with cohesin and TOP2B at active promoters to facilitate DSBR and NER when triggered by the chemotherapy drug mitomycin C (Chatzinikolaou et al, 2023). Supporting a direct role of CTCF in the DDR, *CTCF* is a tumor suppressor gene that is mutationally constrained in the germline, but frequently mutated in cancer (Hilmi et al, 2017; Lang et al, 2017; Segueni & Noordermeer, 2022; Valverde de Morales et al, 2023). Hence, both components of the loop extrusion process—the loop extruders (SMC complexes) and boundary elements (CTCF)—contribute not only to shaping the 3D genome but also, through their redeployment by the DDR, to maintaining its integrity.

### Genome-wide chromatin changes during DSBR

Newer studies have investigated mostly local changes to the genome upon DNA damage, such as histone modifications or positioning (Zavala et al, 2014; García-Nieto et al, 2017; Yancoskie et al, 2024) or protein–protein interactions between architectural and repair factors (Le May et al, 2012; Boetefuer et al, 2018; Lake et al, 2022). Direct comparisons between the DDR and chromatin structure have mostly mapped damage positions to constitutive (i.e., identified from undamaged cells) chromatin features (Heilbrun et al, 2021; Jiang et al, 2021; Akköse & Adebali, 2023;

Wu et al, 2024; Yancoskie et al, 2024). Critically, these studies fail to account for chromatin structure dynamicity, which is already high by default (i.e., before genotoxic stress). Although TAD boundaries tend to be shared not only across cell populations but even across species (Sefer, 2022; James et al, 2024), loop extrusion is highly transient (Gabriele et al, 2022; Mach et al, 2022). This transient nature and high dynamicity warrant the real-time tracking of chromatin structure alongside damage formation and repair. Studies that have done so entail either DSBs or UV-induced damage.

DSBs arise from a wide array of genotoxic insults and cellular processes. These include ionizing radiation, DNA-alkylating agents, unresolved SSBs, UV radiation, replication fork collision, meiotic recombination, antibody diversification in immune cells, and even the repair of other lesion types. To complicate their mapping, several of these processes simultaneously generate additional types of lesions, for example, reactive oxygen species. DSBs are resolved principally by homologous recombination (HR) and the error-prone, nonhomologous end-joining (NHEJ) pathway (Fig 2). Their high tendency to interfere with gene function and form chromosomal aberrations, especially when repaired by NHEJ, makes them highly hazardous and hence associated with aging and disease (reviewed by Gorbunova and Seluanov [2016] and Caldecott [2022]). Consequently, DSBs are one of the most well-studied types of damage.

Because DSBs arise from diverse, overlapping sources, they are often studied in isolation using artificial systems that control where and how often they form. The eight-cutter restriction enzyme AsiSI fused to the estrogen receptor hormone–binding domain induces, upon 4-hydroxytamoxifen treatment, only about a 100 euchromatic DSBs per cell, simplifying the tracking of chromatin structure changes at individual DSBs (Iacovoni et al, 2010). Work in DIvA cells (for *D*SB *i*nducible *vi*a *A*siSI) has generated crucial insights about how DSBR uses and changes large-scale chromatin architecture. DSBR foci are membraneless condensates formed by the accumulation of repair proteins at DSB sites (Arnould et al, 2021; de Luca et al, 2024). Phosphorylation (by ATM kinase) of H2A histone family member X ($\gamma$H2AX), a marker of such foci, spreads until the preexisting boundaries of the DSB-containing TADs (Collins et al, 2020; Arnould et al, 2021), as do the DSBR proteins (de Luca et al, 2024). Conversely, CTCF-bound sites are enriched at $\gamma$H2AX foci (Natale et al, 2017; Ochs et al, 2019). These observations build on previous reports that bound CTCF enhances DSBR (Han et al, 2017; Lang et al, 2017; Natale et al, 2017; Ochs et al, 2019). In support, high-level (i.e., shared across many sub-TADs) TAD boundaries across a variety of cell types show both DSB accumulation and enhanced repair (Chen et al, 2024). DSBs also tend to accumulate at chromatin loop anchors, which are considered fragile sites because of high local transcription and topoisomerase activity (Canela et al, 2017; Gothe et al, 2019). Furthermore, frequently interacting regions of the genome—local hotspots of contact probability on Hi-C maps—tend to coincide with DSB damage sites and to be enriched for repair proteins (Sobhy et al, 2019). These examples are consistent with the notion that repair uses the preexisting chromatin structure to bolster its efficiency. Independently of (preceding) repair, DSB damage formation is also accompanied by a stabilization of existing chromatin structure that is proposed to limit the spread of further damage (Ochs et al, 2019).

The DSB response not only uses the preexisting chromatin structure but also induces changes. The chromatin remodeler AT-rich interaction domain 1A (ARID1A) accumulates at DSBs. Here, it helps to recruit CTCF and the cohesin subunit RAD21 not only to maintain preexisting TADs but also to promote repair-favoring chromatin loop formation (Bakr et al, 2024). In DIvA cells, DSBs in active genes tend to cluster together (Aymard et al, 2017). At a larger scale, damaged TADs also cluster into their own sub-compartment, termed the D compartment (for DSB-induced), within Compartment A (Collins et al, 2020; Arnould et al, 2021, 2023) (Fig 4A). The clustering of these D compartments is thought to boost repair efficiency by multiple mechanisms: concentrating DDR machinery into the same nuclear space; looping DDR genes—which undergo elevated transcription after damage—from undamaged TADs into this same space; and bringing DSB ends close together to be rejoined. The possibility of enhanced repair comes at the cost of compromising genome integrity: having many DSBs from multiple chromosomes in close proximity increases the likelihood that some ends will be misjoined, potentially resulting in translocations. However, the strengthening of TAD boundaries and depletion of chromatin contacts observed at the damage sites may limit how often such translocations occur (Arnould et al, 2023).

The increase in TAD insulation during DSBR is observed in further systems. AsiSI-induced DSBs in mouse embryonic fibroblasts are accompanied by TAD boundary reinforcement, actin-dependent DSB clustering, strengthening of intrachromatin compartment interactions (A with A, B with B), and a slight shift from closed (Compartment B) to open (Compartment A) chromatin (Zagelbaum et al, 2023). Ionizing radiation–induced DSBs in human lymphoblasts and fibroblasts are accompanied by TAD boundary reinforcement and CTCF-mediated loop strengthening (Sanders et al, 2020). Furthermore, proteins concentrated at nuclear speckles, which are involved in the maintenance of intra-compartment interactions (Fig 1), enhance HR-mediated DSBR (Matsui et al, 2020).

In contrast to these cases where chromatin compartmentalization, TADs, and loops are maintained or even strengthened, neurons from patients with Alzheimer's disease, which exhibit genomes overloaded with DSBs (Thadathil et al, 2021), show no change in chromatin compartmentalization. Instead, they display weakened TAD and loop strength, with loop loss linked to the down-regulation of neural development genes (Dileep et al, 2023). Neurons from Alzheimer's patients have reduced CTCF binding at many DNA repair genes, with variable effects on gene expression (Patel et al, 2023). Neurons, as terminally differentiated cells, rely principally on NHEJ to repair DSBs because they lack sister chromatids, leading to the hypothesis that DSB-induced chromatin changes differ between postmitotic and dividing cells (Dileep et al, 2023). It is conceivable that such a breakdown of chromatin structure would accompany the overwhelming of DNA repair systems through a high damage burden, such as that encountered during disease or treatment with DNA-damaging agents. In support, cells treated with the DSB-inducing anthracycline anticancer drug doxorubicin display chromatin compartment switching and a loss of TAD boundaries (Wang et al, 2023b). In contrast, breast

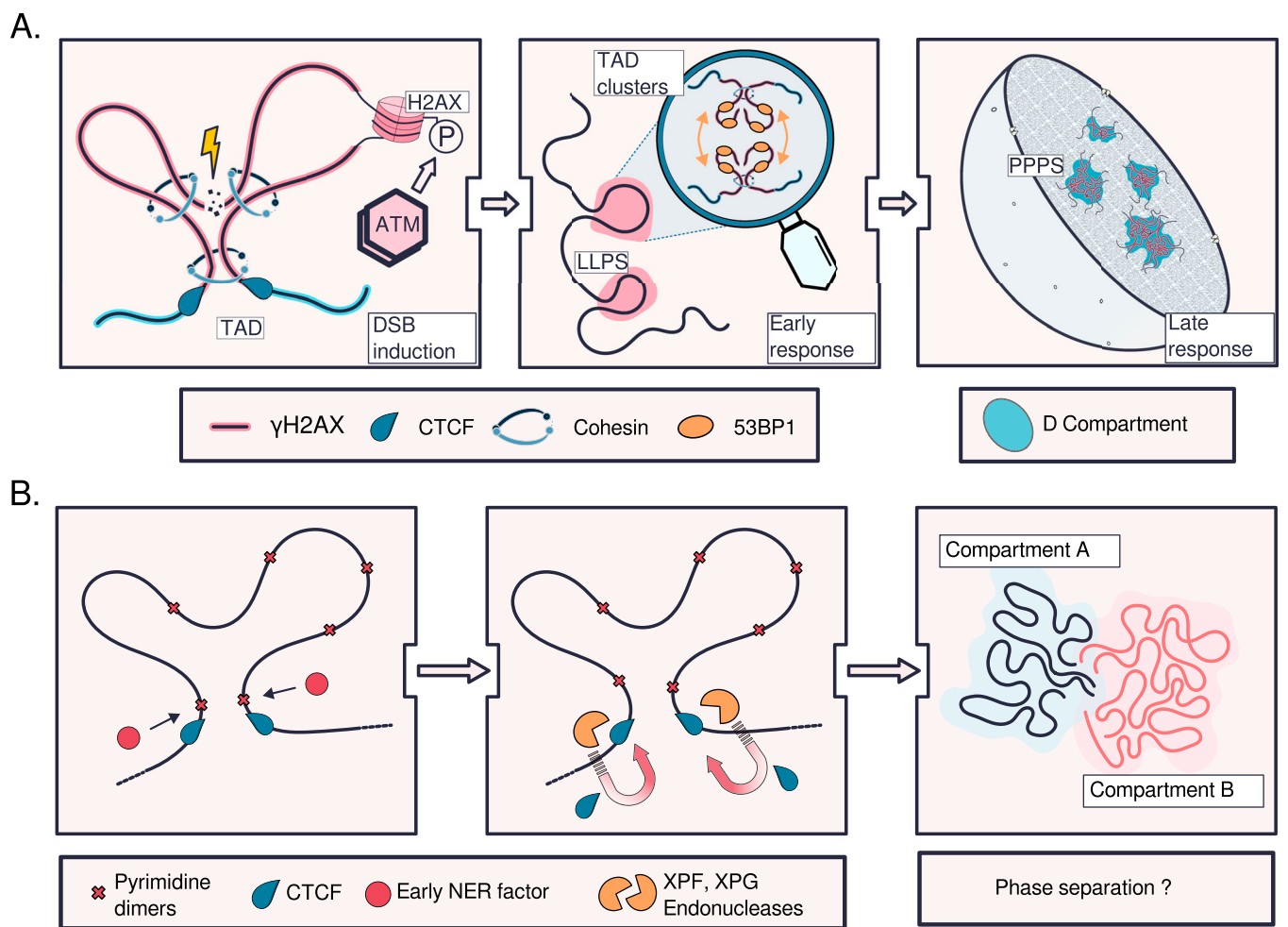

**Figure 4. Reshaping of chromatin domains and compartments during double-strand break repair (DSBR) and nucleotide excision repair (NER).**
**(A)** During the DSB response, one-sided loop extrusion originating at the DSB facilitates the rapid spread and assembly of repair foci, marked by γH2AX (phosphorylation of H2A histone family member X), across preexisting topologically associating domains (TADs) (Collins et al, 2020; Arnould et al, 2021, 2023; de Luca et al, 2024). ATM- and cohesin-dependent intra-TAD contacts are elevated, and inter-TAD contacts are depleted, in damaged compared with undamaged TADs. Key DSBR protein 53BP1 binds damaged TADs, causes them to form liquid condensates leading to clustering of damaged TADs, and spreads across multiple TADs with the help of cohesin (Ochs et al, 2019; Arnould et al, 2023). Clustering initially proceeds through liquid–liquid phase separation (LLPS), which entails weak, transient interactions between chromatin-associated factors and drives accumulation of hundreds of thousands of repair factors at damage sites (Ng et al, 2022; de Luca et al, 2024). As the repair response matures, more stable polymer–polymer phase separation (PPPS) takes over, perhaps allowing the repair machinery to remain anchored to the chromatin (Arnould et al, 2023). **(B)** NER, like DSBR, is accompanied by increased CCCTC-binding factor (CTCF)–mediated chromatin looping, intra-TAD contacts, and TAD boundary strength (Kaya & Adebali, 2025). Early NER factors appear to initiate repair at preexisting TAD boundaries, where high local chromatin accessibility facilitates their recruitment. Subsequently, terminal NER proteins XPF and XPG can recruit CTCF (Le May et al, 2012), potentially explaining the increased TAD boundary strength observed after UV exposure, because TAD insulation positively correlates with CTCF binding. Through an unknown mechanism perhaps including phase separation, compartments also rearrange. Intracompartment interactions, especially among Compartment A, increase, whereas intercompartment (A, B) interactions decrease. The increased compartmentalization across multiple scales of genome architecture may favor repair efficiency as it does during DSBR.

cancer cells that have acquired resistance to carboplatin, a platinum-based anticancer drug that engages both HR and NER, exhibit chromatin structure buildup rather than breakdown compared with carboplatin-sensitive cells, including the formation of new TAD boundaries and chromatin loops (Dozmorov et al, 2023).

Increased chromatin looping also appears to boost DSBR efficiency when the DSBs have been induced by intrinsic cellular processes. For example, the class switch recombination (CSR) and variable, diversity, and end-joining (V(D)J) processes use NHEJ to drive immunoglobulin diversification. At the large and topologically insulated *Igh* (immunoglobulin heavy) locus in mouse B

lymphocytes undergoing V(D)J, formation of very long-range chromatin loops maximizes recombination events of V(D)J segments (Montefiori et al, 2016). Chromatin loop extrusion promotes the synapsis of DSBs induced by activation-induced cytidine deaminase (AID) during CSR (Shen et al, 2021). Mismatches from AID deamination of cytosine to uracil are excised, leaving abasic sites that convert to SSBs and then to DSBs. Active transcription within the murine *Igh* locus creates a novel TAD boundary that aids the DSB synapsis step and thus CSR (Costea et al, 2023).

In summary, findings both from engineered systems—using endonucleases to precisely control DSB location and

frequency—and from natural systems that more accurately reflect cellular exposure to endogenous and exogenous agents (such as ionizing radiation and meiotic or immunoglobulin recombination) are consistent with a model in which the 3D genome "locks up" after damage induction, with the DDR confined to the reinforced boundaries of preexisting TADs, which exhibit increased CTCF-mediated looping. Building on these findings from DSBR, we next explore parallels with NER by summarizing studies involving NER-activating DNA lesions, especially UV-induced photoproducts.

## Genome organization during NER

UV-B radiation and UV-C radiation principally induce cyclobutane pyrimidine dimers (CPDs) and pyrimidine (6-4) pyrimidone pho-toproducts (6-4 PPs). Both CPDs and 6-4 PPs are cleared by the NER pathway. Where and how often they form are largely dictated not only by the underlying sequence composition (Zavala et al, 2014; García-Nieto et al, 2017; Heilbrun et al, 2021) but also by distance from the nuclear center (García-Nieto et al, 2017; Perez et al, 2021; Akköse & Adebali, 2023), nucleosome positioning (Gale et al, 1987; Zavala et al, 2014; Brown et al, 2018; Pich et al, 2018), and the conformational changes to the DNA helix induced by protein binding (Mao et al, 2018; Sivapragasam et al, 2021; Elliott et al, 2023; Yancoskie et al, 2024). Upon UV radiation, chromatin undergoes both nucleosome-stabilizing and nucleosome-destabilizing restructuring (summarized in Apelt et al [2021]). These antago-nistic events are posited to facilitate DNA damage recognition through the global-genome NER (GG-NER) subpathway by pro-viding a scaffold for repair factors while permitting access to repair, but not transcription, machinery (Sabarinathan et al, 2016; Apelt et al, 2021). GG-NER relies on the damage-sensing protein DDB2 to detect CPDs, and on XPC to transfer CPDs and all other substrates to TFIIH for damage verification. XPC interacts preferentially with methylated and deacetylated nucleosomes, whereas all downstream GG-NER factors prefer open chromatin (Bidon et al, 2018; Kusakabe et al, 2022). Upon completion of GG-NER, nucleosomes are repositioned to their predamage configu-ration to ensure faithful continuation of gene expression (Apelt et al, 2021).

Consistent with this nucleosome-resolution "access–repair–restore" model (Smerdon, 1991), time-resolved Hi-C data from UV-damaged cells show rapid and drastic, yet transient, reconfiguration of the 3D genome that positions damaged sites for NER access (Kaya & Adebali, 2025). Within the 1st h after UV exposure, compartmen-talization increases and TAD boundaries strengthen, matching the reinforcement of preexisting chromatin domains captured during DSBR (Collins et al, 2020; Sanders et al, 2020; Arnould et al, 2021, 2023). This 3D reorganization is thought to optimize the proximity of damage-prone regions to repair factors, facilitating efficient local damage scanning without disrupting broader chromatin archi-tecture. As repair progresses, the genome gradually reverts to a conformation similar to its predamage state, signaling NER com-pletion (Kaya & Adebali, 2025).

NER activity beyond the DNA damage recognition step is strongly positively correlated with higher order chromatin ac-cessibility (Adar et al, 2016; Sabarinathan et al, 2016; Mao et al, 2020; Jiang et al, 2021; Anderson et al, 2024; Kaya & Adebali, 2025).

NER prioritizes active, Compartment A chromatin at the nuclear center over inactive, Compartment B chromatin at the nuclear periphery (Akköse & Adebali, 2023; Wu et al, 2024). Because TC-NER operates exclusively within actively transcribed regions, part of this Compartment A preference likely reflects TC-NER activity in addition to general chromatin accessibility—although TC-NER contributes less substantially than GG-NER to total NER activity, with GG-NER responsible for most excision events (Nakazawa et al, 2010; Adar et al, 2016). Disentangling its effect would require TC-NER–deficient models, controlled inhibition of transcription (to eliminate stalled polymerase signal upon which TC-NER initiation relies), or strand-specific measurement of damage or repair to detect the strand asymmetry characteristic of TC-NER activity (e.g., Adar et al, 2016; Sabarinathan et al, 2016). Beyond this TC-NER contribution, proposed models linking ac-cessibility to NER efficiency include the preassembly of repair machinery at highly accessible regions to remove damage soon after its formation (Jiang et al, 2021), including at TAD boundaries (Kaya & Adebali, 2025). Indeed, UV-triggered NER activity is elevated at TAD boundaries and is especially pro-nounced in euchromatic, shorter, and well-insulated TADs with a strong A compartment profile compared with hetero-chromatic, longer TADs (Kaya & Adebali, 2025). Similarly, TC-NER of damage from the fungal carcinogen aflatoxin is elevated at preexisting TAD boundaries and chromatin loop anchors (Wu et al, 2024).

The only known published study tracking 3D genome changes in a time-course setting during NER has uncovered notable simi-larities with the DSB response at both the compartment and the TAD level (Kaya & Adebali, 2025) (Fig 4B). We next discuss the implications of this parallelism.

## Impact of chromatin restructuring on the DDR

The studies discussed are consistent with a model in which the chromatin structure changes captured during DSBR and NER are not passive side effects but instead actively contribute to the DDR. The general trend across both pathways is a strengthening of TAD boundaries upon damage and an increased recruitment of chromatin structure players like CTCF and cohesin, which may themselves influence the repair process (Fig 4). Strikingly, CTCF-bound sites and chromatin loop anchors are frequently damaged or mutated in cancer (Katainen et al, 2015; Kaiser & Semple, 2018). Whereas CTCF binding generally suppresses CPD formation (through helical distortion) (Sivapragasam et al, 2021), strong CTCF motifs and CTCF occupancy are associated with higher rates of DSB formation (Canela et al, 2017; Gothe et al, 2019; Raimer Young et al, 2024). Strong CTCF-binding motifs are fragile sites because of their proneness to form unstable secondary structures (Raimer Young et al, 2024). CTCF physically interacts with TOP2B, which relieves the torsional stress created by loop extrusion by cleaving DNA at loop anchors, thus creating DSBs (Gothe et al, 2019). However, offsetting this high DSB formation, CTCF occupancy stimulates DSBR (Hilmi et al, 2017; Lang et al, 2017; Natale et al, 2017; Ochs et al, 2019), in contrast to its inhibitory effect on NER at CPDs (Sivapragasam et al, 2021). In a third DDR pathway example, putative preexisting chromatin loop anchors—inferred from CTCF, cohesin, and

histone mark ChIP-seq datasets—exhibit a marked reduction in X-ray–induced oxidative damage load (Poetsch et al, 2018)—although it is unclear whether this reduction is attributable to protection against damage formation, enhanced base excision repair, or both. From comparison of damage maps with static (i.e., from unchallenged cells) Hi-C maps, oxidative damage accumulation and base excision repair appear to be confined within CTCF-mediated chromatin loops (Scala et al, 2022).

The increased compartmentalization observed during both NER and DSBR and the elevated repair activity at and within CTCF-delimited spaces support the notion that chromatin compartments may help rather than hinder repair, perhaps by dividing the genome into more manageable search spaces to detect damage (speculated in Collins et al [2020] and Arnould et al [2021]) or, if TAD changes persist, by limiting further damage through the prevention of chromosomal translocations (Sanders et al, 2020). If compartments were instead inhibiting DNA repair, then the acquisition of CTCF-binding affinity-reducing mutations might be beneficial—for example, by allowing the machinery to progress further than it otherwise would.

Although the aforementioned observations across multiple high-throughput mapping studies are consistent with an active role of 3D genome reorganization in DNA repair, it is important to acknowledge that many of these findings remain correlative. The DDR is not merely a collection of repair pathways but a broad signaling network encompassing cell cycling checkpoint activation, replication control, and transcriptional reprogramming (Rouse & Jackson, 2002). As such, transcription and other chromatin-altering processes such as cell cycling or DNA replication are tightly intertwined with the DDR and can themselves remodel chromatin after damage, confounding causal interpretation. DNA damage frequently disrupts gene transcription—although the latter tends to occur on a longer timescale than the early repair phase—through mechanisms such as RNA polymerase II stalling at photolesions or damage in transcription factor motifs leading to differential binding. In other cases, specific genes become transcriptionally up-regulated; for instance, the pronounced chromatin reshuffling observed upon DSB induction in the DIvA system is accompanied by up-regulation of DDR genes (Arnould et al, 2023). Future studies that combine synchronized damage induction with real-time measurements of transcription and chromatin conformation, or that otherwise control for these cofounders—for instance, by inhibiting cell cycling or acutely suppressing transcription before damage induction—will help to disentangle such interdependencies.

# DDR and 3D genome interconnection in genetic disease

The seminal findings from chromatin responding to DSBs and to NER-triggering lesions leave no doubt that the fate of DNA damage is intricately linked to the higher order organization of the genome. However, many unanswered questions, such as the extent to which other DNA repair pathways also use or change chromatin structure, remain to be investigated, with important implications for genotoxic studies in humans.

Apart from DSBs and NER-triggering events, to the authors' knowledge, the only other published higher order chromatin structure studies pertaining to the DDR involve cells in which damage has already been converted to mutations, such as during disease (e.g., Akdemir et al, 2020; Anderson et al, 2024). Investigating whether other damage types and repair pathways (Fig 2) induce similar changes can provide new insights into potential therapeutic strategies, especially should targeting the underlying chromatin structure influence multiple repair pathways simultaneously. Such an approach could enhance the efficacy of DNA-damaging and other replication stress–inducing drugs used to treat cancer and autoimmune disorders, which have previously been challenged by the redundancy across different repair pathways. However, this strategy may need to be carefully titrated to minimize harmful effects on gene transcription and other nuclear processes in nondiseased cells. One such promising strategy involves the use of anthracyclines, chemotherapy drugs that induce DSBs by inhibiting topoisomerases. Recent work on anthracyclines shows that they cause drastic reorganization of long-range chromatin contacts by disrupting CTCF binding at loop anchors (Wang et al, 2023b; Tan et al, 2024 Preprint), although the mechanistic link to their DNA-damaging potency has yet to be explored.

In addition to supplementing preexisting strategies that administer DNA-damaging drugs to destroy diseased cells, knowledge of the interrelatedness of the DDR and chromatin structure will aid efforts to counter genetic diseases stemming from mutations in chromatin architecture proteins. For example, cohesinopathies, caused by mutations in the cohesin pathway, are an area of active research. The *NIPBL* gene, which encodes the cohesin loader Nipped-B–like protein, is the most frequently mutated gene in Cornelia de Lange syndrome (CdLS), a severe autosomal dominant cohesinopathy (Krantz et al, 2004). CdLS patients exhibit increased sensitivity to DNA damage (Vrouwe et al, 2007). CdLS *NIPBL* variants, along with somatic variants of cohesin genes, are linked to genomic instability and various cancers (Pallotta et al, 2022). NIPBL interacts with the chromatin remodeler BRD4, influencing DDR protein 53BP1 to regulate HR during DSBR (Olley et al, 2018, 2021). *NIPBL* mutations common to CdLS disrupt loop extrusion in vitro, consistent with the idea that cohesin mutations hinder the recruitment of DDR proteins to damage sites (Panarotto et al, 2022). In support, in mouse fibroblasts, the NIPBL regulator PRR12 colocalizes with NIPBL to DNA damage sites via cohesin, and its depletion leads to increased DNA damage (Nguyen et al, 2025). Meanwhile, CdLS-implicated mutations and deletions in the cohesin subunit *RAD21*, when introduced into cell lines, substantially impair DSBR after ionizing radiation (Deardorff et al, 2012). Roberts syndrome (RBS), a cohesinopathy caused by mutations in the cohesin regulator *ESCO2* (establishment of cohesion 2), shares hallmark phenotypes with DNA repair deficiency diseases such as ataxia telangiectasia, Bloom syndrome, Fanconi anemia, and Cockayne syndrome (Mfarej & Skibbens, 2020). RBS-derived cells, like those from CdLS, exhibit hypersensitivity to DNA damage and impaired DSBR (McKay et al, 2019). Mouse models with RBS mutations in *Esco2* show early up-regulation of the DDR during embryonic development (Strasser et al, 2024).

CTCF-related disorders are autosomal dominant because of germline intolerance of *CTCF* mutations. They typically involve missense variants in the DNA-binding domain of *CTCF* and lead to intellectual disabilities and developmental delays (Valverde de Morales et al, 2023). Although CTCF depletions in mammalian cell lines—in keeping with the role of CTCF in promoting HR of DSBs—result in hypersensitivity to DNA damage (Hilmi et al, 2017; Lang et al, 2017), thus far, no attempt has been made to study the DDR in such patients. Similarly, although *CTCF* mutations and reduced CTCF binding have been implicated in neuro-developmental disorders wherein neurons acquire high levels of DSBs, such as Alzheimer's disease, their impact on the DDR has not been investigated (Dileep et al, 2023; Patel et al, 2023; Zhang et al, 2024). Instead, the role of CTCF in promoting the DDR has primarily been explored in cancer studies, wherein *CTCF* is considered a tumor suppressor gene (Segueni & Noordermeer, 2022). These observations highlight the need for further research on the relationship between the DDR and the 3D genome in genetic diseases.

## Conclusions and future perspectives

As high-throughput chromatin conformation capture techniques such as Hi-C become increasingly available, we are poised to gain critical insights into how the highly dynamic, adaptable chromatin landscape influences the efficiency and fidelity of DNA repair processes. An intriguing question that remains is how DNA damage triggers genome reorganization. It is unclear how, how often, and for which types of damaging agents the damage itself acts as a direct physical trigger (e.g., helix-distorting lesions) versus indirectly reconfiguring chromatin through, for instance, recruitment of chromatin remodelers, including histone chaperones. Damage-triggered PTMs also likely contribute, as PTMs influence the phase separation processes underlying chromatin compartmentalization (Wang et al, 2019, 2023b; Li et al, 2023). In addition, precursors to damage, such as the binding of transcription factors that alter DNA susceptibility to damage, could contribute. For instance, CTCF-bound sites, which are prone to DSB formation, are susceptible to forming unstable secondary DNA structures (Raimer Young et al, 2024), potentially triggering further restructuring. Answering these mechanistic questions, perhaps by combining local damage induction with advanced imaging techniques, will deepen our understanding of how physical forces and biochemical signaling converge to maintain genome integrity.

There is high motivation to conduct further research on DNA repair in the 3D genome. The high prevalence of CTCF- and cohesin-binding mutations in cancers, especially melanomas, incentivizes clinical studies (Katainen et al, 2015; Poulos et al, 2016; Otlu et al, 2023). From a more basic research angle, the much older evolutionary age of chromatin loop extrusion compared with, for example, the cellular defense system of V(D)J recombination (reviewed in Peters [2021]) impels research about how newer transactions reuse ancient preexisting processes and structures. Loop extrusion, which is prevalent even in gene-poor heterochromatin and is not a universal requirement for transcription,

may have initially evolved to direct topoisomerases to keep the interphase genome unentangled. The resulting unknotted genome may have integrated loop extrusion to facilitate the same nuclear processes, namely, transcription and repair, that it enabled in the first place (through maintenance of an unentangled state) (Dyson et al, 2021; Polovnikov et al, 2023; Hildebrand et al, 2024). A related field is the study of how viral infections change the host genome, with decreased compartmentalization contrasting the strengthening of compartment and domain boundaries observed during NER and DSBR (Wang et al, 2023a; Chiariello et al, 2024; Venu et al, 2024).

Finally, we expect such studies to contribute to current efforts to understand the complex genotypes and phenotypes stemming from genetic diseases such as cohesinopathies because of deficiencies in chromatin architecture proteins. Understanding the 3D genome response during disease will be essential as we advance into the era of personalized medicine. As the race to develop more effective DNA-damaging therapeutics against diseases intensifies, gaining insight into how our genome maintains its integrity under stress becomes more crucial than ever.

## Acknowledgements

The authors thank SJ Sturla for her inspiring comments. Generative AI was used to improve the clarity and conciseness of the writing; all conceptual framing, interpretation of the literature, and conclusions are the authors' own. O Adebali discloses support for this work from TUBITAK (The Scientific and Technological Research Institution of Türkiye), 2232 International Fellowship for Outstanding Researchers Program (grant ID 118C320), Young Scientist Grant (BAGEP) by Science Academy (Türkiye), and European Molecular Biology Organization Installation Grant. Research in the laboratory of H Naegeli is supported by the Swiss National Science Foundation grant 219340. MN Yancoskie discloses support from the Novartis Foundation for Medical-Biological Research, Young Investigator Grant (#24C191). Funding for open access charge was provided by the Swiss National Science Foundation grant 219340.

### Author Contributions

VO Kaya: visualization and writing—original draft, review, and editing.
O Adebali: conceptualization and writing—review and editing.
H Naegeli: conceptualization and writing—review and editing.
MN Yancoskie: conceptualization and writing—original draft, review, and editing.

### Conflict of Interest Statement

The authors declare that they have no conflict of interest.

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
