## [Reviewer comments · Life Science Alliance]

Conserved 3D genome reorganization during DNA repair

Veysel Kaya, Ogun Adebali, Hanspeter Nägeli, and Michelle Yancoskie

DOI: <https://doi.org/10.26508/lsa.202503498>

Corresponding author(s): Michelle Yancoskie, University of Zurich

Review Timeline:	Submission Date:	2025-08-28
	Editorial Decision:	2025-10-17
	Revision Received:	2025-11-03
	Editorial Decision:	2025-11-21
	Revision Received:	2025-11-24
	Accepted:	2025-11-26

Scientific Editor: Tim Fessenden

Transaction Report:

October 17, 2025

Re: Life Science Alliance manuscript #LSA-2025-03498-T

Dr Michelle N Yancoskie
University of Zurich
Institute of Pharmacology and Toxicology
Winterthurerstrasse 260
Zurich 8057
Switzerland

Dear Dr. Yancoskie,

Thank you for submitting your manuscript entitled "Conserved 3D genome reorganization during DNA repair" to Life Science Alliance. The manuscript was assessed by expert reviewers, whose comments are appended to this letter. We sincerely regret the long delay in returning these reviews to you, due to reviewer availability.

As you will see, both reviewers were enthusiastic about the timely and important subject of this work as well as the conceptual advance proposing genome reorganization itself as a regulator of the DNA damage response. Both reviewers made several suggestions to improve this work, which we invite you to consider.

To upload the revised version of your manuscript, please log in to your account: <https://lsa.msubmit.net/cgi-bin/main.plex>
You will be guided to complete the submission of your revised manuscript and to fill in all necessary information. Please get in touch in case you do not know or remember your login name.

-- Summary blurb (enter in submission system): A short text summarizing in a single sentence the study (max. 200 characters including spaces). This text is used in conjunction with the titles of papers, hence should be informative and complementary to the title. It should describe the context and significance of the work for a general readership; it should be written in the present tense and refer to the work in the third person. Author names should not be mentioned.

B. MANUSCRIPT ORGANIZATION AND FORMATTING:

Thank you for this interesting contribution to Life Science Alliance. We are looking forward to receiving your revised manuscript.

Sincerely,

Reviewer #1 (Comments to the Authors (Required)):

The authors present a review on DNA repair in the 3D genome, with a focus here on genome organization into compartments, TADs, loops, and structural proteins that shape the 3D genome. There is limited data on this topic in the literature, but they summarize the available evidence obtained primarily for double strand break (DSB) repair and UV damage repair by nucleotide excision pathways. One problem in this field is that the two events - DNA repair at the sequence level and 3D genome configuration - have rarely been analyzed in the same experimental system and with the same biological samples. The authors conclude that the 3D genome is initially "locked up" after damage to allow more efficient processing of the damage within specific regions. However, it remains unknown how this occurs mechanistically. Overall, the review is useful. I have a few suggestions for improvement.

1) The second paragraph on page 5 beginning with "The DDR is further linked to the 3D genome" Is confusing. Information is summarized in bits and pieces from the literature, but there is no unifying conclusion.

2) How do you disentangle specific 3D features of the genome, such as euchromatin or A-compartments, and TCR, which is a process known for a long time to occur in active genomic regions?

3) Figure 3:

Is the model presented in Figure 3, the bodyguard hypothesis, really relevant and true? We know that nuclear DNA in peripherally located heterochromatin is more AT-rich, and that simply explains the higher frequency of CPDs, a type of UV lesion that prefers TT sequences as target. What independent evidence would support such a model?

Minor:

Page 5, top:

What do the authors mean by "non-genomic links?"

Reviewer #2 (Comments to the Authors (Required)):

This is a timely review into the relationship between the 3D architecture of the genome and genome stability. This field has advanced significantly in recent years - and therefore a review and discussion of the current knowledge is of utmost relevance and importance. The authors introduce the major concepts and knowledge regarding the organization of the genome, and proceed to describe recent findings into the relationship between DNA damage formation and repair and this organization. I find the review to be generally well organized and clear. I have only minor suggestions and comments:

1) Perhaps there is room to explain (in broad terms) the basic methods for measuring 3D interactions and Damage-mapping - or referring to relevant reviews on the techniques.

2) It is important to include a note about the correlative nature of many of the observations - and that many of the processes also affect the transcriptional program in cells - that in turn could affect repair/organization indirectly.

3) In the section on challenges in damage and repair mapping:

- It would be important to explicitly mention the challenge the damages form in different position in every cell in the population - before offering single-cell as a solution. The DiVa system was one way to overcome this challenge.
- In this respect, de Luca et al 2024 measured repair protein recruitment at single-cell resolution - but not damage. To my knowledge measurement of damage by single cell sequencing hasn't been done yet.
- In general - this section referred primarily to mapping of UV damages-and should also included techniques for mapping DSB.

4) In the section on genome wide chromatin changes during DSB (p.7) the authors state: "Despite the complexity in their formation, DSBs have been studied in isolation in the context of the chromatin landscape using artificial systems to control where and how abundantly they form". I would soften this statement as the authors themselves later describe additional research that did not use the DiVA system.

5) In the section on Genome organization during NER (p.9) - the first sentence should state UVB/C - given that the authors

previously mentioned UVA which forms different major adducts.

6) In the section on impact of chromatin restructuring on the DDR (p.11) - it is important to note that the paper of Sivapragasam et al, 2021 showed primarily repression of CPD formation but also inhibition of repair (in contrast to DSB).

7) I find that figure 2 could be improved to better reflect the damages.

- Instead of having the base names (T, U, G) appear as squares outside the helix - I would place them between them. That would allow to show the mispairing of G/T or G/U as aberrant.
- I would place the U on the right side of the helix, at the end of the BER arrow.
- NER deals with pyrimidine dimers and additional bulky lesions. I would add those.
- Pyrimidine dimers are technically not "bulky" adducts - but are helix distorting.

Point-by-point response to the reviewers

We thank both reviewers for their thoughtful comments that have substantially improved the quality of our manuscript. We provide both a tracked-changes and a clean version of the revised manuscript through the submission portal.

Reviewer #1:

1) The second paragraph on page 5 beginning with "The DDR is further linked to the 3D genome" Is confusing. Information is summarized in bits and pieces from the literature, but there is no unifying conclusion.

Response: We thank the reviewer for highlighting the lack of cohesion in this paragraph, which we have restructured to improve synthesis. The revised text first summarizes findings involving SMC complexes, followed by CTCF-related observations, and emphasizes that these examples span multiple DDR pathways and involve both core components of loop extrusion.

2) How do you disentangle specific 3D features of the genome, such as euchromatin or A-compartments, and TCR, which is a process known for a long time to occur in active genomic regions?

Response: We have restructured the third paragraph of "Genome organization during NER" to address this important caveat and now note that NER enrichment in Compartment A likely reflects both transcription-coupled and architecture-dependent effects. Disentangling these influences would require TC-NER-deficient models, controlled inhibition of transcription (to eliminate the stalled polymerase signal upon which TC-NER initiation relies), or strand-specific measurement of damage or repair (e.g. Adar et al, 2016, Sabarinathan et al, 2016).

3) Figure 3: Is the model presented in Figure 3, the bodyguard hypothesis, really relevant and true? We know that nuclear DNA in peripherally located heterochromatin is more AT-rich, and that simply explains the higher frequency of CPDs, a type of UV lesion that prefers TT sequences as target. What independent evidence would support such a model?

Response: We include the bodyguard hypothesis to introduce the historical development of since-refined ideas linking nuclear architecture to genome stability. We agree that sequence composition, as a major determinant of CPD and 6-4 PP formation frequency, constitutes an important caveat meriting discussion and have now expanded the main text and Figure 3 legend to clarify: The genome-wide damage-mapping studies we cited in support of this hypothesis have found, through nucleotide-frequency simulations or otherwise statistically controlled normalization, that peripheral heterochromatin experiences greater UV-induced damage formation than expected

from base composition alone (Akköse and Adebali, 2023; Perez et al, 2022; García-Nieto et al, 2017).

Minor points:

4) Page 5, top: What do the authors mean by "non-genomic links?"

Response: We acknowledge that “non-genomic” is an ambiguous term that could be misinterpreted as referring to non-DNA processes rather than to evidence from non-sequencing-based approaches (e.g. biochemical or microscopy studies). To clarify, we have revised the section title to “Mechanistic links between chromatin structure and DNA repair from non-sequencing approaches” and added a qualifying statement.

Reviewer #2:

Minor points:

1) Perhaps there is room to explain (in broad terms) the basic methods for measuring 3D interactions and Damage-mapping - or referring to relevant reviews on the techniques.

Response: We thank the reviewer for this suggestion and have now added a brief explanation of 3C-based methods, re-citing relevant reviews, at the start of the main section “Adaptability of the 3D genome.” In addition, we have expanded the section “Challenges of damage and repair mapping” to include concise descriptions of representative damage-mapping techniques—covering both UV-induced photolesions and, as noted in the reviewer’s third comment, double-strand breaks, as well as other lesion types.

2) It is important to include a note about the correlative nature of many of the observations - and that many of the processes also affect the transcriptional program in cells - that in turn could affect repair/organization indirectly.

Response: We thank the reviewer for highlighting this important caveat, which we now address in an additional paragraph concluding the section “Impact of chromatin restructuring on the DDR.” We acknowledge, in addition to transcription, other potential cofounders of DDR-associated genome reorganization such as replication and cell cycling—both intimately intertwined with the DDR—and mention experimental strategies that could help isolate repair-specific events.

3) In the section on challenges in damage and repair mapping:

• It would be important to explicitly mention the challenge the damages form in different position in every cell in the population - before offering single-cell as a solution. The DiVa system was one way to overcome this challenge.

• In this respect, de Luca et al 2024 measured repair protein recruitment at single-cell resolution - but not damage. To my knowledge measurement of damage by single cell sequencing hasn't been done yet.

• In general - this section referred primarily to mapping of UV damages-and should also included techniques for mapping DSB.

Response: We appreciate these clarifications and have removed the de Luca reference as an example of single-cell damage mapping. As suggested, we now introduce the discussion of single-cell approaches by first highlighting the fundamental challenge posted by the stochastic nature of damage formation, referencing the DiVA system (discussed later in the text) as an example of how this issue can be addressed. Finally, we have expanded this section to include techniques for mapping DSBs, in addition to those detecting UV-induced and other lesions (see also our response to Comment 1).

4) In the section on genome wide chromatin changes during DSB (p.7) the authors state: "Despite the complexity in their formation, DSBs have been studied in isolation in the context of the chromatin landscape using artificial systems to control where and how abundantly they form". I would soften this statement as the authors themselves later describe additional research that did not use the DiVA system.

Response: We now clarify that artificial systems such as DiVA represent a common, but not exclusive, approach to studying DSBs in the context of genome architecture.

5) In the section on Genome organization during NER (p.9) - the first sentence should state UVB/C - given that the authors previously mentioned UVA which forms different major adducts.

Response: We have corrected this oversimplification as recommended by specifying the UV subtype.

6) In the section on impact of chromatin restructuring on the DDR (p.11) - it is important to note that the paper of Sivapragasam et al, 2021 showed primarily repression of CPD formation but also inhibition of repair (in contrast to DSB).

Response: We thank the reviewer for this important clarification and have revised the text to highlight this contrast between the two repair pathways.

7) I find that figure 2 could be improved to better reflect the damages.

• Instead of having the base names (T, U, G) appear as squares outside the helix - I would place them between them. That would allow to show the mispairing of G/T or G/U as aberrant.

• I would place the U on the right side of the helix, at the end of the BER arrow.

• NER deals with pyrimidine dimers and additional bulky lesions. I would add those.

• Pyrimidine dimers are technically not "bulky" adducts - but are helix distorting.

We thank the reviewer for these helpful suggestions to improve Figure 2. The base symbols (T, U, G) are now positioned between the DNA strands to illustrate mismatching and base modification. Regarding NER, we have expanded the figure legend to mention additional representative lesion types—such as intrastrand crosslinks, DNA–protein crosslinks, bulky aromatic adducts, and cyclopurine lesions—while retaining only two visual examples (photoproducts and interstrand crosslinks) to maintain clarity and balance of the figure layout. Finally, we have replaced the term “bulky” with “helix-distorting” when describing pyrimidine dimers, both in the legend and in the main text.

November 21, 2025

RE: Life Science Alliance Manuscript #LSA-2025-03498-TR

Dr. Michelle N Yancoskie
University of Zurich
Institute of Veterinary Pharmacology and Toxicology
Winterthurerstrasse 260
Zurich 8057
Switzerland

Dear Dr. Yancoskie,

Thank you for submitting your revised manuscript entitled "Conserved 3D genome reorganization during DNA repair". Only Reviewer 1 was available to re-review this work, and as you will see below they are satisfied. We would be happy to publish your paper in Life Science Alliance pending final revisions necessary to meet our formatting guidelines.

- Please be sure that the authorship listing and order is correct.
- Please add the X and Bluesky handles of your host institute/organization, as well as your own and/or one of the authors in our system.
- Please consult our manuscript preparation guidelines <https://www.life-science-alliance.org/manuscript-prep> and make sure your manuscript sections are in the correct order.
- The contributions selected for Ogün Adebali do not qualify them for authorship. Please either update the contributions in our system and in the Author Contributions section of the manuscript, or let us know if the author needs to be removed (and potentially added to the acknowledgment section).
- Please add callouts for Figure 1A-D to your main manuscript text .

LSA now encourages authors to provide a 30-60 second video where the study is briefly explained. We will use these videos on social media to promote the published paper and the presenting author (for examples, see <https://docs.google.com/document/d/1-UWCfbE4pGcDdcgzcmiuJl2XMBJnxKYeqRvLLrLSo8s/edit?usp=sharing>). Corresponding or first-authors are welcome to submit the video. Please submit only one video per manuscript. The video can be emailed to contact@life-science-alliance.org

A. FINAL FILES:

B. MANUSCRIPT ORGANIZATION AND FORMATTING:

Thank you for your attention to these final processing requirements. Please revise and format the manuscript and upload materials as soon as you are able.

Sincerely,

Reviewer #1 (Comments to the Authors (Required)):

The authors have revised the manuscript according to the reviewer comments. I am satisfied with the revisions.

November 26, 2025

RE: Life Science Alliance Manuscript #LSA-2025-03498-TRR

Dr. Michelle N Yancoskie
University of Zurich
Institute of Veterinary Pharmacology and Toxicology
Winterthurerstrasse 260
Zurich 8057
Switzerland

Dear Dr. Yancoskie,

Thank you for submitting your Review entitled "Conserved 3D genome reorganization during DNA repair". It is a pleasure to let you know that your manuscript is now accepted for publication in Life Science Alliance. Congratulations on this interesting work.

DISTRIBUTION OF MATERIALS:

Again, congratulations on a very nice paper. I hope you found the review process to be constructive and are pleased with how the manuscript was handled editorially. We look forward to future exciting submissions from your lab.

Sincerely,
